# Accelerating AutoDock Vina with GPUs

**DOI:** 10.3390/molecules27093041

**Published:** 2022-05-09

**Authors:** Shidi Tang, Ruiqi Chen, Mengru Lin, Qingde Lin, Yanxiang Zhu, Ji Ding, Haifeng Hu, Ming Ling, Jiansheng Wu

**Affiliations:** 1School of Geographic and Biological Information, Nanjing University of Posts and Telecommunications, Nanjing 210023, China; 1020173019@njupt.edu.cn (S.T.); 1021173601@njupt.edu.cn (J.D.); 2Smart Health Big Data Analysis and Location Services Engineering Research Center of Jiangsu Province, Nanjing University of Posts and Telecommunications, Nanjing 210023, China; 3VeriMake Research, Nanjing Renmian Integrated Circuit Technology Co., Ltd., Nanjing 210088, China; rickychen@verimake.com (R.C.); mengru.lin@verimake.com (M.L.); zhuyanxiang@verimake.com (Y.Z.); 4National ASIC System Engineering Technology Research Center, Southeast University, Nanjing 210096, China; 220206029@seu.edu.cn (Q.L.); trio@seu.edu.cn (M.L.); 5School of Telecommunication and Information Engineering, Nanjing University of Posts and Telecommunications, Nanjing 210023, China; huhf@njupt.edu.cn

**Keywords:** AutoDock Vina, Vina-GPU, OpenCL, GPU

## Abstract

AutoDock Vina is one of the most popular molecular docking tools. In the latest benchmark CASF-2016 for comparative assessment of scoring functions, AutoDock Vina won the best docking power among all the docking tools. Modern drug discovery is facing a common scenario of large virtual screening of drug hits from huge compound databases. Due to the seriality characteristic of the AutoDock Vina algorithm, there is no successful report on its parallel acceleration with GPUs. Current acceleration of AutoDock Vina typically relies on the stack of computing power as well as the allocation of resource and tasks, such as the VirtualFlow platform. The vast resource expenditure and the high access threshold of users will greatly limit the popularity of AutoDock Vina and the flexibility of its usage in modern drug discovery. In this work, we proposed a new method, Vina-GPU, for accelerating AutoDock Vina with GPUs, which is greatly needed for reducing the investment for large virtual screens and also for wider application in large-scale virtual screening on personal computers, station servers or cloud computing, etc. Our proposed method is based on a modified Monte Carlo using simulating annealing AI algorithm. It greatly raises the number of initial random conformations and reduces the search depth of each thread. Moreover, a classic optimizer named BFGS is adopted to optimize the ligand conformations during the docking progress, before a heterogeneous OpenCL implementation was developed to realize its parallel acceleration leveraging thousands of GPU cores. Large benchmark tests show that Vina-GPU reaches an average of 21-fold and a maximum of 50-fold docking acceleration against the original AutoDock Vina while ensuring their comparable docking accuracy, indicating its potential for pushing the popularization of AutoDock Vina in large virtual screens.

## 1. Introduction

Molecular docking studies how two or more molecular structures (e.g., drug and target) fit together. Molecular docking analysis has become one of the most common methods for modern drug discovery [1]. It allows the prediction of molecular interactions where a protein and a ligand can be inducted to fit together in the bound state. Additionally, molecular docking tools provide an efficient and cheap method, in the early stage of drug design, for the identification of leading compounds and their binding affinities [2,3,4]. Among all molecule docking tools, the AutoDock suite is the most popular, which consists of various tools including AutoDock4 [5], AutoDock Vina [6], AutoDock Vina 1.2.0 [7], AutoDock FR [8], AutoDock Crank Pep [9,10], AutoDock-GPU [11,12] etc. AutoDock Vina is usually recommended as the first-line tool in the implementation of molecular docking due to its docking speed and accuracy [13]. Moreover, it shows the best docking power in the last benchmark CASF-2016 for comparative assessment of scoring functions [14] and the best scoring power under the comparison with 10 docking programs on a diverse protein—ligand complex sets [15]. AutoDock Vina uses a Monte-Carlo iterated local search method, which comprises iterations of sampling, scoring and optimization. First, an initial random conformation is sampled for a given ligand, which is represented by its position, orientation and torsion (POT). Then, its position, orientation or torsion is randomly mutated with a disturbance. Finally, the affinity is evaluated for the binding pose of a ligand and a protein. In AutoDock Vina, the binding affinity is calculated by a scoring function which describes the sum of the intermolecular energy (ligand–receptor) and the intramolecular energy (ligand–ligand). Moreover, the conformation is optimized with a Broyden–Fletcher–Goldfarb–Shanno (BFGS) method [16] that considers the gradients of the scoring function. These gradients can guide the ligand to achieve a better conformation with a lower docking score. In addition, a metropolis acceptance rule [17], which relies on the difference between the docking score of the initial conformation and that of the optimized conformation, is arranged to decide whether the optimized conformation can be accepted or not. The accepted conformation will be recorded as the initial conformation and further optimized in the next iterations. As all we know, the Monte-Carlo based iterated local search method in AutoDock Vina is highly serialized because the ongoing iteration depends on the previous outputs.

Preceding virtual screens pipeline typically operates only on a scale of $10^{6}$∼$10^{7}$ compound molecules. Such scale of compounds will heavily descend the capability and increase the failure risk of modern drug discovery. Fortunately, the whole chemical space of drug-like molecules has been estimated to reach more than $10^{60}$ [18]. The scale of compounds in virtual screens is vital since the more candidate compounds there are to be screened, the lower rate of failure and the more favorable quality of leading compounds can be reached [19]. Hence, the virtual screens on huge compound databases are urgent for identifying excellent drug candidates in modern drug discovery. However, original virtual screening with AutoDock Vina on huge databases is very slow, which cannot meet the need for modern drug discovery. Therefore, an acceleration of AutoDock Vina has become a central problem in current virtual screens of drug hits from huge compound databases. Till now, there have been several attempts for the acceleration of AutoDock Vina in large virtual screens [20,21,22]. For instance, VirtualFlow provides a drug discovery platform that speeds up AutoDock Vina in virtual screening of an ultra-large database with more than 1.4 billion molecules by leveraging over 160,000 CPUs [20]. Such huge resource investment and expenditure, as well as the high entry threshold for users, seriously weaken the popularization of AutoDock Vina and the flexibility of customer’s usage (such as a self-defined target and small molecule dataset). Due to the overall serial design of the AutoDock Vina algorithm, its parallelization mostly depends on the stacking of computing powers as well as the allocation of resources and tasks under such a common scenario that faces large virtual screens in modern drug discovery. A reduction of computational resource investment and user access threshold will advance the broad spread of AutoDock Vina in large virtual screening for modern drug discovery.

Graphic processing unit (GPU) is a powerful parallel programmable processor with thousands of computing cores that provide a tremendous computational performance. GPU has become an integral part of mainstream computing systems due to the high price–performance ratio and the ease of developing an implementation with well-established standards such as Compute Unified Device Architecture (CUDA) and Open Computing Language (OpenCL). GPU has been applied to accelerate molecular docking in several tools [11,12,23,24,25,26,27,28,29,30,31,32]. For example, AutoDock-GPU provides an OpenCL implementation of AutoDock4 to exploit both GPU and CPU parallel architectures. By exploring three levels of parallelism (runs, individual, fine-grained tasks) on the Lamarckian Genetic Algorithm (LGA) algorithm, AutoDock-GPU achieves the maximum of 350-fold acceleration of total runtime against the single-threaded CPU implementation [11]. Recently, an attempt has been made into the GPU parallel acceleration of AutoDock Vina where the Viking method tried to rewrite the pose search stage of AutoDock Vina on GPU [30]. Till now, no positive acceleration result has been reported. The reasons for its deficiency probably involve the following three points. Firstly, the Monte Carlo-based optimization process in AutoDock Vina is the most time-consuming (typically more than 90%) and highly dependent whose next iteration relies on the previous outputs. Secondly, each ligand file was presented as a heterogeneous tree structure whose nodes are traversed recursively. Thirdly, the CUDA architecture on NVIDIA GPU cards limits its cross-platform portability.

In this work, we proposed an efficient parallel method, namely Vina-GPU, to accelerate the classic algorithm—Monte-Carlo based simulated annealing as well as the optimizer—BFGS in AutoDock Vina with GPUs. First, Vina-GPU applies a large-scale parallelism on the Monte-Carlo based iterated docking threads and significantly reduces the search depth in each thread. Second, a heterogeneous OpenCL implementation was efficiently deployed on Vina-GPU by converting the heterogeneous tree structure into a list structure whose nodes are stored in the traversed order. The two implementations ensure that Vina-GPU can leverage thousands of computational cores on GPU and achieve a large-scale parallelization and acceleration, and realizes the cross-platform portability on both CPUs and GPUs. Large benchmark tests show that Vina-GPU reaches an average of 21.66× and a maximum of 50.80× speed-up on NVIDIA Geforce RTX 3090 against the original AutoDock Vina on a 20-threaded CPU while ensuring their comparable docking accuracy. To further enlarge its potential of pushing the popularity of AutoDock Vina in large virtual screening, more efforts have been taken, as follows. First, we fitted a heuristic function for automatically determining the most important hyperparameter (*search_depth*) on the basis of large testing experiments in order to lower the usage threshold. Second, we developed a user-friendly graphical user interface (GUI) for a convenient operation of Vina-GPU. Third, we enable the implementation of Vina-GPU to be built on Windows, Linux and macOS, ensuring their usability on personal computers, station servers and cloud computations etc. The code and tool of Vina-GPU are freely available at https://github.com/DeltaGroupNJUPT/Vina-GPU (accessed on 27 March 2022) or http://www.noveldelta.com/Vina_GPU (accessed on 27 March 2022).

## 2. Methodology

The heterogeneous OpenCL implementation of Vina-GPU is depicted in Figure 1, which consists of a host part (on CPU) and a device part (on GPU). The host part is mainly in charge of the preparation and post-refinement of the conformations. The device part focuses on the acceleration of the most time-consuming Monte Carlo-iterated local search method by enlarging the scale of parallelism as well as reducing the number of iterations.

### 2.1. Host Part

The host part consists of two sections (see Figure 1). The first section includes four operations, which are the file reading, the OpenCL setup, the data preparation and the device memory allocation, and all operations are implemented for the input to the device part. Specifically, the file-reading operation is to read the ligand and protein files in .pdbqt format, and the OpenCL setup operation is to setup the OpenCL environment (platform, device, context, queue, program and kernels). Furthermore, the host part prepares all the required data, including grid cache (for calculating the energy of a conformation), random maps (for generating probability random numbers) and random initial conformations (for the Monte Carlo-based method to start from). The data is then re-organized to load in the device memory according to how it is accessed (read-only or read-write). The read-only grid cache, random maps and random conformations are allocated in the constant device memory while the read–write of best conformations returned by the device part is allocated in the global device memory. Such kind of memory management could efficiently boost the speed of reading and writing on GPU. The second section includes multiple operations after the device part. All the best conformations returned from the device part are clustered and sorted in the container by their docking scores. The best 20 conformations will be concretely refined and optimized before generating the final output ligand files.

In the data preparation operation, AutoDock Vina treats each conformation as a heterogeneous tree structure whose nodes are stored with its frame information and a pointer to its children node. Each node is traversed by a depth-first search policy to calculates the conformation energy in a recursive process. However, in Vina-GPU, the OpenCL standard cannot support any recursion in kernels because the allocation of stack space for thousands of threads is too expensive. In addition, various ligands would generate different heterogeneous trees that are not suitable for the OpenCL implementation. Therefore, we transformed the heterogeneous tree structure into a list type (see Figure 2), each node of which is stored in line with their traversed order. These nodes can be traversed simply by the order of the node list. In addition, a children map was created to denote the relationship among these nodes. For example, the node 0 has two children-nodes (the node 1 and the node 4) and so the row 0 has two “T”s (indicating “True”) in the 1st and 4th column (Figure 2). Thus, the recursive traverse of the heterogeneous tree can be converted into an iterative traverse of the node list and children map, which fits the OpenCL standard.

### 2.2. Device Part

On the device part, the allocated constant memory (highlighted in orange in Figure 1) is assigned for the initialization and the calculation during the reduced-step Monte-Carlo iterated local search processes (highlighted in green in Figure 1) and the final best conformations are stored in global memory (highlighted in gold in Figure 1).

Vina-GPU enables thousands of reduced-steps iterated local search processes running concurrently within the GPU computational cores. We denote each reduced-step iterated local search process as a docking thread. Within each thread, an OpenCL work item is assigned to a randomly initialized conformation C, which can be represented by its position, orientation and torsion (POT):(1)$$C=\{x,y,z,a,b,c,d,\psi_{1},\psi_{2},\ldots ,\psi_{N_{rot}}\}$$
where x,y,z correspond to the position of the conformation in a pre-determined searching space; a,b,c,d denote its orientation as a rigid body in the quaternion form; $\psi_{1},\psi_{2},\ldots ,\psi_{N_{rot}}$ represent torsions of $N_{rot}$ rotatable bonds. Then, each conformation C is to be randomly mutated in one of its POT with the uniform distribution. The conformation will be continuously evaluated with a scoring function that quantifies the potential energy of the binding pose. Generally, the potential energy *e* is calculated with the sum of intermolecular energy and intramolecular energy:(2)$$e=e_{inter}+e_{intra}$$
where $e_{inter}$ represents the interaction energy between the ligand and the receptor, and it is calculated using trilinear interpolation that approximates the energy of each atom pair by looking up the grid cache; and $e_{intra}$ indicates the interaction energy of the pairwise atoms within the ligand. Considering that both $e_{inter}$ and $e_{intra}$ are related to the binding pose, the scoring function SF can be denoted as a function of POT variables:(3)$$\mathrm{SF}=f(x,y,z,a,b,c,d,\psi_{1},\psi_{2},\ldots ,\psi_{N_{rot}})$$

After the energy evaluation, a Broyden–Fletcher–Goldfarb–Shanno (BFGS) [16] optimization is applied to update the ligand conformation by minimizing of the scoring function SF. Essentially, the BFGS method is to substitute the Hessian matrix $H\in R^{\left(7+N_{rot}\right)\times \left(7+N_{rot}\right)}$ with an approximate matrix $B\in R^{\left(7+N_{rot}\right)\times \left(7+N_{rot}\right)}$ whose inverse matrix $B^{-1}$ is iteratively updated by the first-order derivatives $\nabla \mathrm{SF}\left(C\right)\in R^{\left(7+N_{r}ot\right)}$. $B_{k+1}^{-1}$ in the (k+1)th iteration can be calculated by
(4)$$\begin{array}{cc} B_{k+1}^{-1}=B_{k}^{-1}\; & +\frac{\left(s_{k}^{T}y_{k}+y_{k}B_{k}^{-1}y_{k}\right)\left(s_{k}s_{k}^{T}\right)}{{\left(s_{k}^{T}y_{k}\right)}^{2}}\\ \; & -\frac{B_{k}^{-1}y_{k}s_{k}^{T}+s_{k}y_{k}^{T}B_{k}^{-1}}{s_{k}^{T}y_{k}} \end{array}$$
where
(5)$$\begin{array}{cc} s_{k} & =-\alpha_{k}B_{k}^{-1}\nabla SF\left(C_{k}\right) \end{array}$$
(6)$$\begin{array}{cc} \alpha_{k} & =argmin\;SF\;\left(C_{k}\;+\alpha p_{k}\right) \end{array}$$
(7)$$\begin{array}{cc} y_{k} & =\nabla SF\left(C_{k}\;+\alpha_{k}p_{k}\right)-\nabla SF\left(C_{k}\right) \end{array}$$
where $B_{0}$ is initiated with identity matrix E and the detailed calculation of ∇SF(C) is described in [6]. Furthermore α ($\alpha_{k}$) means the step size in the direction p ($p_{k}$), along the decrease of the SF value. Next, a metropolis-acceptance criterion is adopted to decide whether to accept the optimized conformation or not, by comparing the energy $e_{0}$ before the mutation and the energy $e_{opt}$ after the optimization. Here, the accept probability *P* is represented by:(8)$$P=\left\{\begin{array}{cc} 1 & \;e_{0}>e_{opt}\;\\ \frac{\exp(e_{0}-e_{opt})}{1.2} & \;e_{0}\leq e_{opt}\; \end{array}\right.$$

This indicates that the accepted conformation is more likely to have a lower energy. Once accepted, the conformation will be further evaluated and optimized by BFGS. Then, the next iteration continues to update the previous optimized conformations until convergence. Finally, all the best conformations found by work items are returned to the host part. Algorithm 1 proposed the pseudocode of our Vina-GPU. In Algorithm 1, Mutation(.) means a random mutation of the POT in a ligand conformation; BFGS(.) represents the BFGS optimization method which is described in Equations (4)–(7); Scoring(.) is the potential energy of a binding pose described in Equations (2) and (3); Metropolis(.) is the metropolis acceptance criterion described in Equation (Equation 8); and Clustering&Sorting(.) is the aggregation and reordering (based on the docking score) of all ligand conformations among all threads.
**Algorithm 1** Vina-GPU method**Input:** random ligand conformations: $\left\{C_{{tmp}_{0}},C_{{tmp}_{1}},\ldots ,C_{{tmp}_{N}}\right\}$**Output:** top *k* ligand conformations $\left\{C_{0},C_{1},\ldots ,C_{k-1}\right\}\in \left\{C_{{tmp}_{0}},C_{{tmp}_{1}},\ldots ,C_{{tmp}_{N}}\right\}$  1: **for all** $C_{tmp_{i}}\;\left(i=0,1,\ldots ,N\right)$ **concurrently do**  2:  **for all**
search_depth=0,1,2,⋯,r
**do**  3:   
$C_{{cand}_{i}}=Mutation\left(C_{{tmp}_{i}}\right)$  4:   
$\left(C_{{cand}_{i}},e_{cand_{i}}\right)=BFGS\left(C_{{cand}_{i}}\right)\&Scoring\left(C_{{cand}_{i}}\right)$  5:   **if** $search\_depth==0\;\|Metropolis\left(e_{cand_{i}},e_{tmp_{i}}\right)$ **then**  6:    $C_{{tmp}_{i}}=C_{{cand}_{i}}$  7:    
$\left(C_{{cand}_{i}},e_{cand_{i}}\right)=BFGS\left(C_{{tmp}_{i}}\right)\&Scoring\left(C_{{tmp}_{i}}\right)$  8:   **end if**  9:  **end for**10: **end for**11: $Clustering\;\&\;Sorting(C_{{tmp}_{0}},C_{{tmp}_{1}},\ldots ,C_{{tmp}_{N}})$12: **return**
$\left\{C_{0},C_{1},\ldots ,C_{k-1}\right\}\in \left\{C_{{tmp}_{0}},C_{{tmp}_{1}},\ldots ,C_{{tmp}_{N}}\right\}$

## 3. Results and Discussion

### 3.1. Experimental Settings

All 140 complexes in the AutoDock-GPU study [12] are assigned as our experimental dataset, which is comprised of 85 complexes from the Astex Diversity Set [33], 35 complexes from CASF-2013 [34], and 20 complexes from the Protein Data Bank [35]. They cover a wide range of ligand complexities and targets properties. Each complex file includes an X-ray structure, an initial random pose of its ligand and the corresponding receptor (in .pdbqt format). We created a config.txt file for each complex (see the example in Appendix A), which involves the center (indicated by $center_{x},center_{y},center_{z}$) and the recommended volume of the docking box (indicated by $size_{x},size_{y},size_{z}$). We classified the 140 complexes into three subsets by their atom sizes $N_{atom}$ (small—5–23 atoms; medium—24–36 atoms; large—37–108 atoms). The details of our experimental data can be seen in Appendix A.

AutoDock Vina was executed on Intel (R) Core (TM) i9-10900K CPU @ 3.7 GHz using Windows 10 Operating System with 64 GB RAM. AutoDock Vina was customized by several configurable arguments, including the center and the volume of searching spaces, the number of CPU cores (*cpu* ) to be utilized and the docking runs (exhaustiveness) etc. The argument exhaustiveness was set to 128 [36], and the argument cpu was set to the maximum value of 20 for taking a full use of the CPU computational power.

Vina-GPU was developed with OpenCL v.3.0 and executed on three different GPUs (Nvidia Geforce GTX 1080Ti, Nvidia Geforce RTX 2080Ti, Nvidia Geforce RTX 3090 (Nvidia Corporation, CA, USA) ) under single-precision floating-point format (FP32). Details are included in Appendix A. In our Vina-GPU, we replaced cpu and exhaustiveness with the number of threads (thread) and the size of searching iterations in each thread (search_depth). These two hyperparameters are of the most importance, and their values are vital to the docking performance of Vina-GPU. For a convenient usage of Vina-GPU, as in AutoDock Vina [6], a heuristic formula was fitted to automatically determine the proper size of search_depth for a given complex. Specifically, a large number of tests were executed on all 140 complexes to examine their docking performance under various sizes of search_depth, where the proper search_depth that guarantees a comparable docking performance was selected. Then, the least squares method was used to fit an empirical formula of the proper search_depth with respect to the $N_{atom}$ (the number of atoms) and $N_{rot}$ (the number of rotatable bonds) in a ligand. The heuristic formula is given as follows,
(9)$$\begin{array}{cc} search\_depth & =max(1,floor(0.24*N_{atom}\\ \; & +0.29*N_{rot}-3.41)) \end{array}$$
where the function floor(∗) gives the largest integer less than or equal to ∗.

### 3.2. Influence of Hyperparameters

We evaluated the influence of the hyperparameters thread (from 100 to 15,000) and search_depth (from 1 to 50) on the docking accuracy (evaluated by docking score and RMSD) as well as the docking runtime of Vina-GPU. The docking score represents the binding affinity between a ligand and a receptor (the lower the score, the better) and the RMSD measures the atom distance difference between an output conformation and the ground truth X-ray structure (again, the lower the better) [6]. An acceptable docking is defined if the least RMSD among all output conformations of Vina-GPU is smaller than 2 Å [13]. Three complexes (5tim, 2bm2 and 1jyq) were randomly selected, which represent various levels of complexities (small, medium and large). The influence of thread and search_depth on the docking score, RMSD and docking runtime are shown in Figure 3 and Figure 4, respectively. All experiments were executed under NVIDIA Geforce RTX 3090 GPU card.

With the increase of thread, the docking score improves and it becomes convergent when the size of thread reaches around 6000 for 2bm2 and 1jyq, and about 1000 for 5tim (Figure 3a). The same trend is also observed on the RMSD performance (Figure 3b), where 2bm2 and 1jyq converge at around 8000, and 5tim fluctuates slightly nearby 2 Å. In Figure 3c, with the raise of thread, the docking runtimes of all three complexes increase slowly. Although it is enough for the small complex 5tim to obtain the best docking accuracy with 1000 thread, the size of thread needs to be set around 8000 for the medium complex 2bm2 and large complex 1jyq. Thus, the size of thread was set to be 8000 for all 140 complexes in this paper.

For the small-complex 5tim, with the increase of search_depth, its docking score, RMSD and docking runtime stay steady. The size of search_depth does not influence the docking results, because Vina-GPU can achieve the best performance with a few search_depth for such a small complex (Figure 4). For the medium complex 2bm2, the docking score and RMSD converge quickly with the raise of search_depth, and the docking runtime increases slowly. This is because a medium complex needs more search_depth to reach the convergence (Figure 4). For the large complex 1jyq, the docking score and RMSD converge slowly with search_depth. The docking runtime for 1jyq increases rapidly with search_depth, because the device runtime for such a large complex 1jyq takes the major part of the total (host + device) docking runtime, increasing search_depth leads to a great expense on the total docking runtime.

### 3.3. Docking Accuracy

We compare the overall docking accuracy of Vina-GPU with AutoDock Vina in terms of the docking score and RMSD performances on all 140 complexes (Figure 5). The color bar encodes the number of atoms in a ligand. For the docking score, most complexes distribute around the diagonal line and fall into the lavender margin of 0.5 kcal/mol difference and their Pearson correlation coefficient of the scores is 0.965 (Figure 5a), which denotes a significant positive correlation. The average docking score of AutoDock Vina and Vina-GPU are −8.9 and −8.7, respectively. These results show that our Vina-GPU achieves the very close docking scores with AutoDock Vina on CPU core.

A docking conformation is typically acceptable when its RMSD difference with the ground truth structure is smaller than 2 Å [12]. In Figure 5b, the red dashed line distinguishes whether a docking conformation is acceptable or not from the RMSD aspect. Figure 5b demonstrates that most complexes fall into the lower left region where both Vina-GPU and AutoDock Vina succeed to obtain the acceptable docking. For Vina-GPU, 107 out of 140 RMSD results are within 2 Å, while 114 out of 140 for AutoDock Vina. The average RMSD of AutoDock Vina and Vina-GPU are 1.5 and 1.7, respectively. These results show that our Vina-GPU achieves the similar docking RMSD with AutoDock Vina. Thus, these findings indicate that Vina-GPU exhibits the comparable docking accuracy with respect to AutoDock Vina on both docking score and RMSD.

### 3.4. Runtime Comparison

The runtime acceleration (Acc) of Vina-GPU against AutoDock Vina is defined by
(10)$$\begin{array}{c} Acc=\frac{t_{vina}}{t_{vina\text{-}gpu}} \end{array}$$
where $t_{vina}$ and $t_{vina\text{-}gpu}$ is the runtime of AutoDock Vina and Vina-GPU, respectively. Figure 6 shows the runtime acceleration (Acc) on various scales of complexity (small—5–23 atoms, medium—24–36 atoms, large—37–108 atoms) and different GPU cards (Nvidia Geforce GTX 1080Ti, Nvidia Geforce RTX 2080Ti, Nvidia Geforce RTX 3090, respectively). The average acceleration is highlighted by a white dot in the center.

As indicated in Figure 6, Vina-GPU achieves the maximal acceleration of 50.80×, as well as the average of 8.84×, 12.70× and 21.66× on the 1080ti, 2080ti and 3090 GPU cards, respectively. The results show that the average acceleration increases along with the complexity of the complex (from small to large) and also raises with higher-end GPU cards (from NVIDIA Geforce GTX 1080ti to NVIDIA Geforce GTX 3090). Figure 7 shows the Acc performance of all 140 complexes along with different $N_{atom}$ and $N_{rot}$. Each bar represents a complex coupling with its corresponding acceleration. As shown in Figure 7, the acceleration varies from 1.03× to 50.80×. The maximal acceleration 50.80× is achieved on the 1xm6 (PDBid) complex under Nvidia Geforce RTX 3090 GPU card.

Among the whole Vina-GPU program, the Monte Carlo-based optimization process is the most time-consuming part (typically more than 90%), which is performed in the “device” part utilizing GPU computational cores. For a better exhibition of the acceleration in the most time-consuming part, we defined the device runtime acceleration $Acc_{d}$ as
(11)$$\begin{array}{c} Acc_{d}=\frac{t_{mc}}{t_{d}} \end{array}$$
where $t_{mc}$ is the Monte Carlo-based optimization part runtime of AutoDock Vina and $t_{d}$ is the device part runtime of Vina-GPU. As shown in Figure 8 and Figure 9, Vina-GPU achieves the maximum of 191.68× and the average of 18.94×, 43.58× and 48.48× acceleration on Nvidia Geforce GTX 1080ti, Nvidia Geforce RTX 2080Ti and Nvidia Geforce RTX 3090 GPU cards, respectively.

### 3.5. Conformation Spaces Analysis

To verify their equivalence in molecular docking, we intend to analyze the full conformation spaces explored by AutoDock Vina and our Vina-GPU. Firstly, we discussed the searching strategy of Vina-GPU and explain why Vina-GPU can achieve a great acceleration on the premise of comparable docking accuracy. Then, we visualized and compared their whole searching of conformation spaces.

Vina-GPU enables thousands of docking threads to run concurrently. These docking threads divide the whole search space into thousands of subspaces, and in each subspace an initial conformation is being optimized. We define the search space that covers all possible conformations as a high-dimensional space $S=\left\{C_{0},C_{1},C_{2},\ldots \right\}$. By dividing S into *n* sub-spaces, we have
(12)$$\begin{array}{c} S=\left\{S_{sub_{0}},S_{sub_{1}},\ldots ,S_{sub_{n}}\right\} \end{array}$$
and each initial conformation belongs to a sub-space
(13)$$\begin{array}{c} C_{i}\in {S_{sub}}_{i}\;\;\;\;\left(i=0,1,2,\ldots \right) \end{array}$$

For each initial conformation $C_{i}$, the corresponding searching space $S_{sub_{i}}$ is much smaller than the whole searching space S. Therefore, we can greatly reduce the searching iterations of each initial conformation in each $S_{sub_{i}}$. By clustering and sorting all the best conformations, Vina-GPU ensures a comparable docking accuracy with original AutoDock Vina.

Then, we detailed a case (PDBid: 2bm2, $N_{a}tom=33,N_{rot}=7$) and visualized their full searching of conformation spaces in Figure 10. AutoDock Vina was executed with the configuration of “cpu=1,exhaustiveness=1 and search_depth=22,365 (default value)”. Vina-GPU was executed under various strategies, where different sizes of thread and search_depth were used. The whole searching spaces (threads×search_depth) were kept almost the same as that of original AutoDock Vina. All conformations searched by AutoDock Vina or our Vina-GPU are indicated as orange or blue dots, respectively. Each conformation is represented by its POT in Cartesian coordinates, where a principal component analysis (PCA) method was used to reduce the dimensionality of orientation and torsion into three. The best conformation is shown by the red star (indicated by an arrow).

As shown in Figure 10a,b, the whole conformation space reached by Vina-GPU or AutoDock Vina is almost the same in their position, orientation or torsion. With the increase of Vina-GPU on its parallel threads and the reduce of search_depth in each thread, these observations stay unchanged (Figure 10c,d). Moreover, the best conformations found by AutoDock Vina or our Vina-GPU are very close to each other. These results demonstrate that our Vina-GPU can achieve comparable docking accuracy with the original AutoDock Vina.

### 3.6. Comparison with the Implementation of Vina-GPU on CPUs

Due to the inherently serial characteristic of the AutoDock Vina algorithm, our Vina-GPU proposed an improved algorithm and then accelerated it with GPUs. For evaluating the contributions of the algorithm improvement and the GPU hardware acceleration separately, we gave out the performance comparison with the implementation of Vina-GPU on CPUs (Figure 11 and Figure 12). The implementation of Vina-GPU on CPUs was executed on Intel (R) Core (TM) i9-10900K CPU @ 3.7 GHz. Both these implementations were executed on all 140 complexes with the same settings of thread (8000) and search_depth. The results of our Vina-GPU on GPUs are identical to those in Figure 6.

For the docking score, most complexes lie around the diagonal line and fall into the lavender margin of a 0.5 kcal/mol difference, only with a few exceptions due to the randomness of Vina-GPU algorithm (Figure 11a). The Pearson correlation coefficient of their docking scores is 0.966 (Figure 11a). The average docking score of Vina-GPU on CPUs and GPUs are −8.6 and −8.7, respectively. The results show that the implementation of Vina-GPU on CPUs achieves comparable docking scores. For the docking RMSD, most complexes gather in the bottom-left region, which indicates that they are acceptable dockings for the implementations of Vina-GPU on CPUs and GPUs (Figure 11b). And for Vina-GPU on CPUs, 104 out of 140 RMSD results are within 2 Å, while 107 out of 140 on GPUs. The average RMSD of Vina-GPU on CPUs and GPUs are 1.8 and 1.7, respectively. The comparable docking scores and RMSD mean that the implementation of Vina-GPU on CPU obtains the almost same docking accuracy. For the docking runtime, the acceleration of the implementation of Vina-GPU on GPUs is higher than that on CPUs, and the latter one only achieves the maximal acceleration of 26.19× and the average of 3.49×, 8,81× and 18.76× on the small, medium and large complexes against the original AutoDock Vina, respectively (Figure 12). These results indicate that our improved algorithm with the implementation on both the CPUs and GPUs gains the comparable docking accuracy, and it is more suitable for the implementation on GPU hardware to achieve higher accelerations.

### 3.7. A Case for Virtual Screening

To show the acceleration effect of our Vina-GPU in implementing real virtual screens of compound databases, a case was detailed on the receptor 1xm6 (PDBid) with the docking of DrugBank [37]. The receptor 1xm6 is the catalytic domain of human phosphodiesterase 4B in complex with (R)-mesopram, and DrugBank is one of the most popular drug databases that contains comprehensive information on drugs and drug targets. A total of 9125 molecules were downloaded from the DrugBank database at https://go.drugbank.com/releases/latest#structures (accessed on 27 March 2022). Both Vina-GPU and AutoDock Vina were executed on the same computer with Intel (R) Core (TM) i9-10900K CPU @ 3.7 GHz and NVIDIA Geforce RTX 3090 GPU card. The exhaustiveness and cpu of AutoDock Vina were set to 128 and 20, respectively. The thread and search_depth of Vina-GPU were set to 8000 and the heuristic value, respectively. Only ~9.44 h was needed to execute the whole docking process by Vina-GPU while ~133.90 h was needed by AutoDock Vina, indicating that the acceleration of 14.18× are achieved by our Vina-GPU. The docking scores of all 9125 molecules on Vina-GPU and AutoDock Vina are shown in Appendix A. We evaluated the similarity of top *i* compounds with the lowest docking scores on AutoDock Vina or our Vina-GPU by Jaccard index [38] as defined by
(14)$$\begin{array}{c} J_{i}=\frac{\left|T_{vina}^{i}\cap T_{vina\text{-}GPU}^{i}\right|}{\left|T_{vina}^{i}\cup T_{vina\text{-}GPU}^{i}\right|} \end{array}$$
where i=15,50,100,200,300, and $T_{vina}^{i}$ and $T_{vina\text{-}GPU}^{i}$ represents subset of top *i* compounds of AutoDock Vina and Vina-GPU, respectively. Table 1 shows that all the Jacard indexes are larger than 0.8, indicating a high similarity of the docking results of Vina-GPU and AutoDock Vina.

Figure 13 shows the comparison of docking scores between Vina-GPU and AutoDock Vina, where most compounds lie around the diagonal line and within the margin (in lavender) of 0.5 kcal/mol difference on the docking score. The Pearson correlation coefficient of their docking scores is 0.981. The average docking socre of AutoDock Vina and Vina-GPU are −7.9 and −7.8, respectively. These results show that our Vina-GPU achieves highly similar docking scores to AutoDock Vina on CPU.

### 3.8. Usage of Vina-GPU

We developed a user-friendly graphic user interface (GUI) instead of the original terminal form. Our GUI can be utilized without installation and is described in Appendix A. In addition, we provided a detailed guideline on how to build and run Vina-GPU on mainstream operating systems (Windows, Linux and MacOS), and it can also ensure the usability of Vina-GPU on personal computers, station servers and cloud computations, etc. (see Appendix A). All source codes and tools of Vina-GPU are freely available at http://www.noveldelta.com/Vina_GPU (accessed on 27 March 2022) or https://github.com/DeltaGroupNJUPT/Vina-GPU (accessed on 27 March 2022).

## 4. Conclusions

In modern drug discovery, huge resource investment and high entry threshold seriously weaken the popularity of AutoDock Vina in large virtual screening from compound databases. To advance the spread of AutoDock Vina in large virtual screens, we proposed a novel method Vina-GPU to speed up AutoDock Vina with GPUs. Vina-GPU obtains a large-scale of parallelism on the classic algorithm—Monte Carlo-based simulated annealing—and greatly reduces the search depth in each iteration. With one of the traditional optimizers, BFGS, Vina-GPU can update the ligand conformation on the consideration of its gradient. Furthermore, a heterogeneous OpenCL implementation of Vina-GPU was efficiently assigned by transforming the heterogeneous tree structure into a list structure whose nodes are visited in the traversed line. Vina-GPU can fully utilize abundant computational GPU cores to reach a large-scale of parallelization and acceleration. Moreover, Vina-GPU can realize cross-platform operation on both CPUs and GPUs. Large benchmarks demonstrate that Vina-GPU achieves an average speed-up of 21-fold and a maximal speed-up of 50-fold on NVIDIA Geforce RTX 3090 over the original AutoDock Vina when keeping their comparable docking accuracy. To further enlarge the popularity of AutoDock Vina in large virtual screens, more efforts were taken, as the follows. A heuristic function was fitted based on large testing experiments to automatically set the most important hyperparameter (search_depth). Moreover, a graphical user interface (GUI) was designed for a convenient usage of Vina-GPU. In addition, an extension of Vina-GPU was provided on Windows, Linux and macOS, and also ensure its usage on personal computers, station servers, and cloud computations, etc. The source codes of Vina-GPU can be freely accessible at https://github.com/DeltaGroupNJUPT/Vina-GPU (accessed on 27 March 2022) or http://www.noveldelta.com/Vina_GPU (accessed on 27 March 2022). In future studies, the following aspects would be taken into consideration for pushing the popularization of AutoDock Vina in large virtual screens. We will further analyze and mend the AutoDock Vina algorithm so that it can obtain a higher acceleration with GPUs. In addition, we will study other mainstream tools in the AutoDock Vina suites and accelerate them with GPUs. Moreover, we will rewrite the AutoDock Vina algorithm to realize its acceleration on FPGA with higher price–performance ratio and more flexibility. Furthermore, we will further consider the influence of the hyperparameters (GPU threads and depths) on model performance for more levels of complexity of compounds or protein targets.

## Figures and Tables

**Figure 1 molecules-27-03041-f001:**
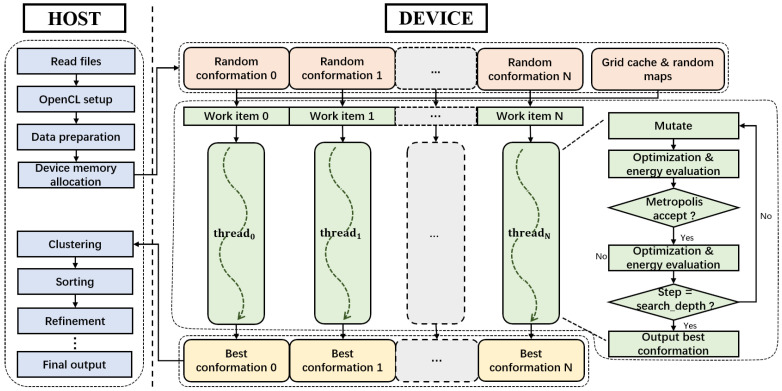
The OpenCL architecture for implementing Vina-GPU, which consists of a host (CPU) and a device (GPU) part of execution. The device part implements thousands of docking threads, each of which is assigned with an OpenCL work item to perform a Monte Carlo-based local search method with largely reduced search iterations.

**Figure 2 molecules-27-03041-f002:**
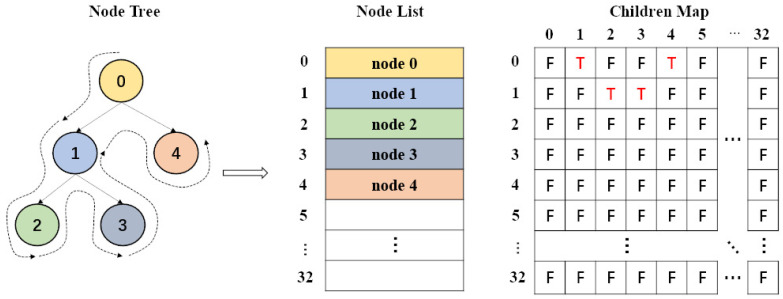
Transformation the original node tree structure into the node-list format. The heterogeneous node tree was reconstructed in its traversed order (depth-first) into the node list where an additional children map was built to reflect the relationship among the nodes. For example, the node 0 has two children-nodes (the node 1 and the node 4) and so the row 0 has two “T”s (indicating “True”) in the 1st and 4th column.

**Figure 3 molecules-27-03041-f003:**
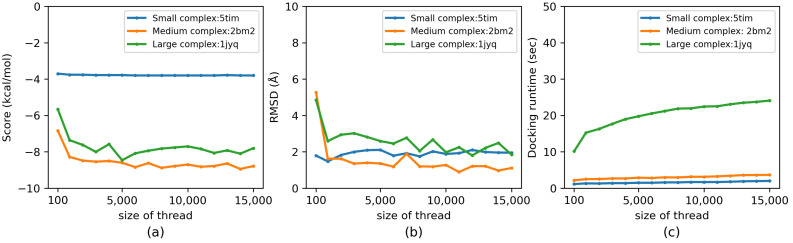
Influence of the size of thread on docking accuracy (score and RMSD) and docking runtime of Vina-GPU in (**a**), (**b**) and (**c**), respectively. Three typical PDB complexes are randomly selected from all 140 complexes which represent small, medium and large ones, respectively (5tim—small, 5 atoms; 2bm2—medium, 33 atoms; 1jyq—large, 60 atoms). All experiments were executed on NVIDIA RTX 3090 GPU card.

**Figure 4 molecules-27-03041-f004:**
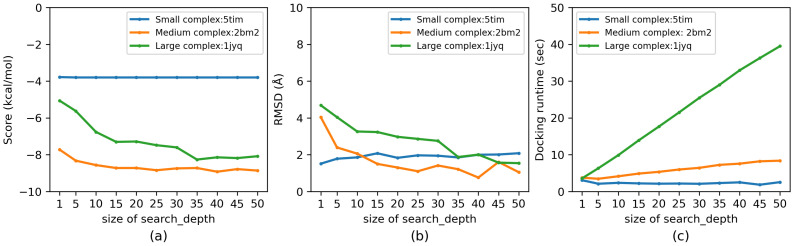
Influence of the size of search_depth on docking accuracy (score and RMSD) and docking runtime of Vina-GPU in (**a**), (**b**) and (**c**), respectively. Three typical PDB complexes were randomly selected from all 140 complexes which represent small, medium and large ones, respectively (5tim—small, 5 atoms; 2bm2—medium, 33 atoms; 1jyq—large, 60 atoms). All experiments were executed on NVIDIA RTX 3090 GPU card.

**Figure 5 molecules-27-03041-f005:**
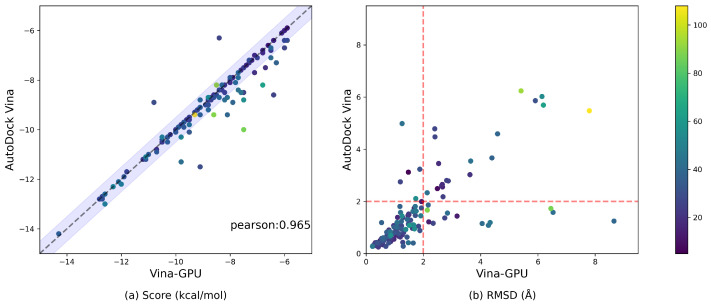
Comparable docking accuracy between AutoDock Vina and our Vina-GPU on all 140 complexes. The color bar encodes the number of atoms in a ligand. A margin of 0.5 kcal/mol difference on the docking score between Vina-GPU and AutoDock is highlighted in lavender in (**a**). The Pearson correlation coefficient of their docking scores is 0.965 (indicated by “pearson”). The RMSD value that indicates an acceptable binding pose (<2 Å) are separated by a red dashed line in (**b**).

**Figure 6 molecules-27-03041-f006:**
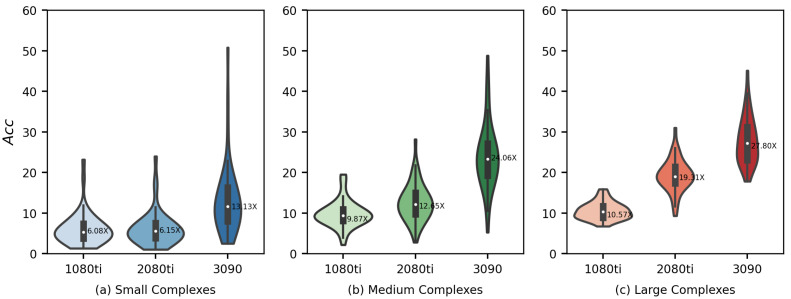
Acceleration of docking time (Acc) of our Vina-GPU against AutoDock Vina on three different GPUs and various scales of complexity (small—5–23 atoms, medium—24–36 atoms, large—37–108 atoms). 1080ti—NVIDIA Geforce GTX 1080ti; 2080ti—Nvidia Geforce RTX 2080Ti; 3090—Nvidia Geforce RTX 3090.

**Figure 7 molecules-27-03041-f007:**
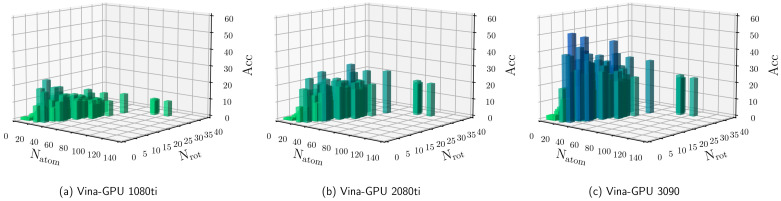
Details for the acceleration of docking time (Acc) of our Vina-GPU against AutoDock Vina on all 140 complexes. The complexity is depicted with their number of atoms ($N_{atom}$) and rotatable bonds ($N_{rot}$). The vertical axis ranges from 0 to 60, and each bar represents a complex coupling with its corresponding acceleration (Acc). Vina-GPU 1080ti, Vina-GPU 2080ti and Vina-GPU 3090 mean that Vina-GPU was executed on Nvidia Geforce GTX 1080ti, Nvidia Geforce RTX 2080Ti and Nvidia Geforce RTX 3090, respectively. The color depth of each bar indicates the value of accelerations (the darker the higher).

**Figure 8 molecules-27-03041-f008:**
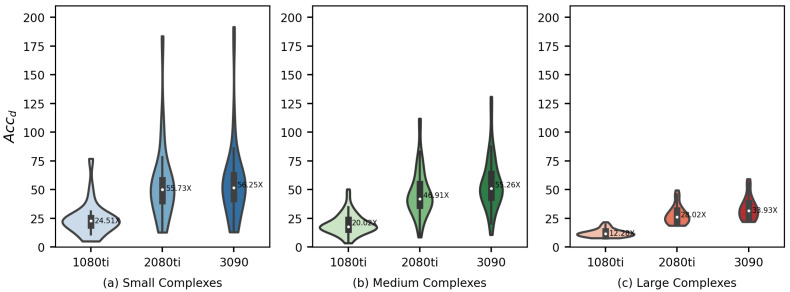
Acceleration ratio on $Acc_{d}$ of our Vina-GPU against AutoDock Vina on three different GPUs and various scales of complexity according to their $N_{atom}$ sizes (small—5–23 atoms, medium—24–36 atoms, large—37–108 atoms). 1080ti—NVIDIA Geforce GTX 1080ti; 2080ti—Nvidia Geforce RTX 2080Ti; 3090—Nvidia Geforce RTX 3090.

**Figure 9 molecules-27-03041-f009:**
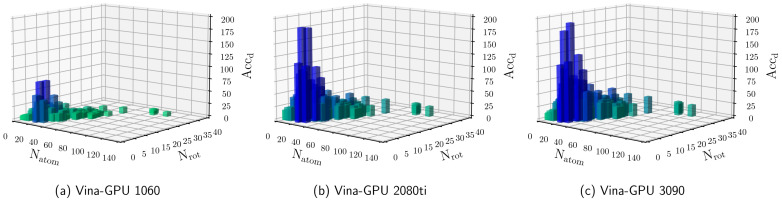
Details for the acceleration of docking time ($Acc_{d}$) of our Vina-GPU against AutoDock Vina on all 140 complexes. The complexity is depicted with their number of atoms ($N_{atom}$) and rotatable bonds ($N_{rot}$). The vertical axis ranges from 0 to 210, and each bar represents a complex coupling with its corresponding acceleration ratio. Vina-GPU 1080ti, Vina-GPU 2080ti and Vina-GPU 3090 mean that Vina-GPU was executed on Nvidia Geforce GTX 1080ti, Nvidia Geforce RTX 2080Ti and Nvidia Geforce RTX 3090, respectively. The color depth of each bar indicates the value of accelerations (the darker the higher).

**Figure 10 molecules-27-03041-f010:**
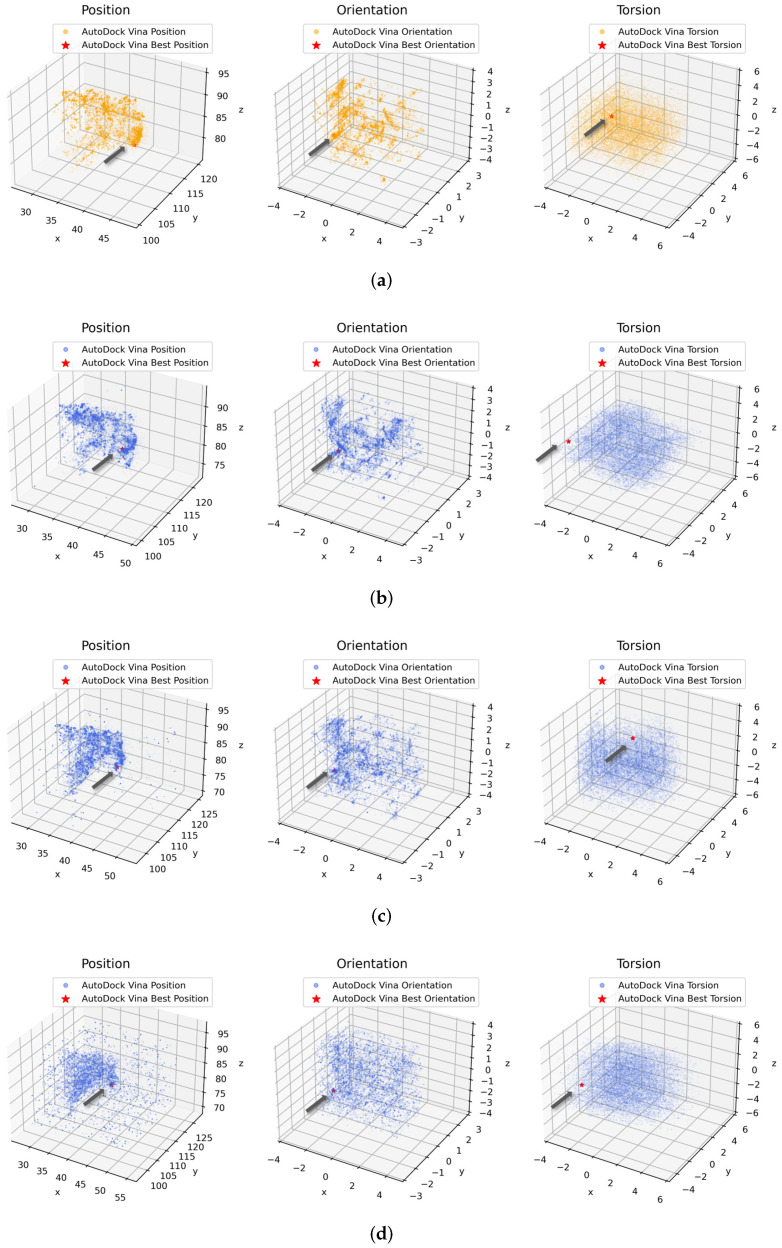
Similar distribution of full conformation spaces (PDBid: 2bm2) explored by AutoDock Vina or our Vina-GPU with various hyperparameter settings. AutoDock Vina (cpu=1, exhaustiveness=1) was executed with one initial conformation and search_depth=22,365 by default. Vina-GPU was performed under different scales of docking threads with various search_depth in each thread. The whole searching iterations (=thread×search_depth) keep almost the same. Each conformation is represented by its position, orientation and torsion (POT), all of which are plotted with orange dots in AutoDock Vina and bule dots in Vina-GPU. The principal component analysis (PCA) method was used to reduce the dimensions of orientation or torsion into three. The best conformations for their final output are highlighted with red stars (pointed by arrows). (**a**) AutoDock Vina: 1 initial conformation with search_depth=22,365. (**b**) Vina-GPU: 10 docking threads with search_depth=2237. (**c**) Vina-GPU: 100 docking threads with search_depth=224. (**d**) Vina-GPU: 1000 docking threads with search_depth=22.

**Figure 11 molecules-27-03041-f011:**
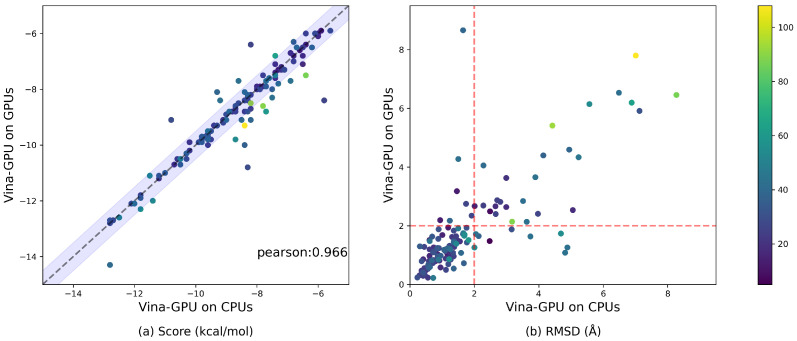
Comparable docking accuracy by the implementation of Vina-GPU on CPUs. All 140 complexes were used, and the color bar encodes the number of atoms in a ligand. A margin of 0.5 kcal/mol difference on the docking score for the implementation of Vina-GPU on CPUs or GPUs is highlighted with lavender in (**a**). The Pearson correlation coefficient of their docking scores is 0.966 (indicated by “pearson”). The RMSD value that indicates an acceptable binding pose (<2 Å) is separated by a red dashed line in (**b**).

**Figure 12 molecules-27-03041-f012:**
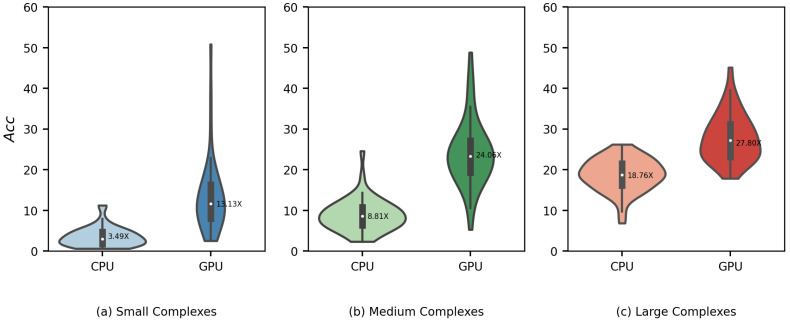
Acceleration of docking time (Acc) of the Vina-GPU implementations on CPUs (indicated by “CPU”) against on GPUs (indicated by “GPU”). All 140 complexes are classified into three sub-datasets with different complexity (small—5–23 atoms, medium—24–36 atoms, large—37–108 atoms). The average acceleration is highlighted with a white dot in the center. The implementation of Vina-GPU on GPUs was executed on Nvidia Geforce RTX 3090 (identical to the results in Figure 6) and that on CPUs was executed on Intel (R) Core (TM) i9-10900K CPU @ 3.7 GHz.

**Figure 13 molecules-27-03041-f013:**
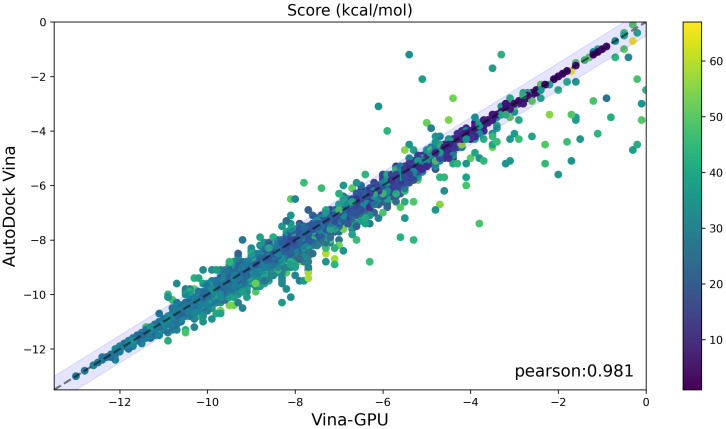
Comparable docking scores between AutoDock Vina and our Vina-GPU on all 9125 compounds from the drugbank dataset. The color bar encodes the number of atoms in one ligand. A margin of 0.5 kcal/mol difference on the docking score between AutoDock and our Vina-GPU is highlighted in lavender. The Pearson correlation coefficient of their docking scores is 0.981 (indicated by “pearson”).

**Table 1 molecules-27-03041-t001:** The Jacard indexes $J_{i}$ on the top *i* subsets of Vina-GPU and AutoDock Vina.

Top *i*	$T_{\mathrm{vina}}^{i}\cap T_{\mathrm{vina}\text{-}\mathrm{GPU}}^{i}$	$T_{\mathrm{vina}}^{i}\cup T_{\mathrm{vina}\text{-}\mathrm{GPU}}^{i}$	Jacard Index
15	14	16	0.875
50	46	54	0.852
100	91	109	0.835
200	187	213	0.878
300	277	323	0.858

## Data Availability

All the source codes, the documentation and the updates related to Vina-GPU have been uploaded to our website http://www.noveldelta.com/Vina_GPU (accessed on 27 March 2022) (in the userguideline section) under an Apache-2.0 license. The 140 complexes described in Section 3.1 are available at https://zenodo.org/record/4031961#.Yags3NBByUk (accessed on 27 March 2022) and the 9125 Drug- Bank molecules used in Section 3.7 are available at https://go.drugbank.com/releases/latest#structures (accessed on 27 March 2022). If there is any problem in reproducing our work, please feel free to contact us.

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
