# Peer review of "Accelerating AutoDock Vina with GPUs"

_molecules, 2022, doi:10.3390/molecules27093041_

Round 1

Reviewer 1 Report

The article presents a new implementation of AutoDock Vina software with GPUs, named Vina-GPU. Vina-GPU is based on a modified Monte-Carlo using simulated annealing algorithm. It raises the number of initial random conformations and reduces the search depth of each thread. Also, other improvements were added as an optimizer of ligand conformations named BFGS, and an OpenCL implementation for parallel acceleration using thousands of GPU cores. The Vina-GPU implementation was tested with a set of 140 protein-ligand complexes classified in small, medium or large, depending on the atoms number. The docking accuracy of Vina-GPU implementation was evaluated using the score function and RMSD parameters respect to AutoDock Vina. Positive correlations were found. The runtime acceleration was analyzed respect to number of atoms and number of rotatable bonds. The implementation was tested in several GPU cards. An interesting test was performed using Vina-GPU on CPUs, obtaining comparable docking scores and docking accuracy, but the runtime acceleration was better using GPUs than CPUs. A case for large virtual screening was evaluated using 9125 molecules from DrugBank. A graphic user interface (GUI) was developed. Vina-GPU can be used in several operating systems (Windows, Linux and MacOS). In general, it is a work very complete and robust. This work has a great interest for the users of molecular docking tools. In addition, it includes the source code and the described tools are freely available, which is an additional benefit to readers and users. Based on before mentioned, I recommend the publication of the manuscript in the Molecules journal, of the MDPI editorial, after some minor improvements.

Minor revisions:

In Abstract section, page 1, line 10, change: “based method” by “based on the method”.

In Abstract section, page 1, line 13, change: “Vina-GPU is based on a modified Monte-Carlo based simulating annealing AI algorithm that greatly raises the number of initial random conformations and reduces the search depth of each thread.” by “Vina-GPU is based on a modified Monte-Carlo using simulating annealing AI algorithm. It greatly raises the number of initial random conformations and reduces the search depth of each thread.”

In Introduction section, page 2, line 47, although the metropolis acceptance rule is a known rule in optimization algorithms, it is suggested to include a reference for metropolis rule.

In Introduction section, page 5, in equations (5)-(7), what information contains the parameters a and ak?

In the Results and Discussion section, page 6, lines 179-190, Tables S1- S3 are cited, but I did not receive the supplementary tables to analyze.

In the Results and Discussion section, page 7, line 207, change “2bm2 and 1jyq converges” by “2bm2 and 1jyq converge”.

In the Results and Discussion section, page 7, lines 219-220, change “the docking score and RMSD converges slowly” by “the docking score and RMSD converge slowly”.

In the Results and Discussion section, page 13, in Figure 10, the information included in the rectangle is not readable. This information should be put in a larger size in each graph of Figure 10.

In the Conclusion section, page16, line 341, change “tranditional optimizer” by “traditional optimizer”.

In the Conclusion section, page16, line 342, change “VINA-GPU can updates” by “VINA-GPU can update”.

In the Conclusion section, page16, lines 361-362, the sentence “A heuristic function was automatically fitted the most important hyperparameter (search_depth) based on large testing experiments.” should be rewritten for better understanding.

In the Conclusion section, page17, line 365, change “station servers and cloud computations etc.” by “station servers, and cloud computations, etc”.

Author Response

Response to Reviewer #1

We would like to thank the reviewer for the valuable insights and suggestions to improve the quality of our manuscript. The following changes have been made to address the Reviewer’s comments, where all the changes are highlighted in yellow in the manuscript.

  1. The Reviewer comments that

The article presents a new implementation of AutoDock Vina software with GPUs, named Vina-GPU. Vina-GPU is based on a modified Monte-Carlo using simulated annealing algorithm. It raises the number of initial random conformations and reduces the search depth of each thread. Also, other improvements were added as an optimizer of ligand conformations named BFGS, and an OpenCL implementation for parallel acceleration using thousands of GPU cores. The Vina-GPU implementation was tested with a set of 140 protein-ligand complexes classified in small, medium or large, depending on the atoms number. The docking accuracy of Vina-GPU implementation was evaluated using the score function and RMSD parameters respect to AutoDock Vina. Positive correlations were found. The runtime acceleration was analyzed respect to number of atoms and number of rotatable bonds. The implementation was tested in several GPU cards. An interesting test was performed using Vina-GPU on CPUs, obtaining comparable docking scores and docking accuracy, but the runtime acceleration was better using GPUs than CPUs. A case for large virtual screening was evaluated using 9125 molecules from DrugBank. A graphic user interface (GUI) was developed. Vina-GPU can be used in several operating systems (Windows, Linux and MacOS). In general, it is a work very complete and robust. This work has a great interest for the users of molecular docking tools. In addition, it includes the source code and the described tools are freely available, which is an additional benefit to readers and users. Based on before mentioned, I recommend the publication of the manuscript in the Molecules journal, of the MDPI editorial, after some minor improvements.

Response:

We thank the reviewer for the nice comment.

  1. The Reviewer comments that

Minor revisions:

  1. In Abstract section, page 1, line 10, change: “based method” by “based on the method”.

Response:

Thanks for your suggestion. In order to eliminate its ambiguity, we changed it to “a new method Vina-GPU”in page 1 line 9 (highlighted in yellow).

  1. In Abstract section, page 1, line 13, change: “Vina-GPU is based on a modified Monte-Carlo based simulating annealing AI algorithm that greatly raises the number of initial random conformations and reduces the search depth of each thread.” by “Vina-GPU is based on a modified Monte-Carlo using simulating annealing AI algorithm. It greatly raises the number of initial random conformations and reduces the search depth of each thread.”

Response:

Thanks for your suggestion. We’ve changed

“Vina-GPU is based on a modified Monte-Carlo based simulating annealing AI algorithm that greatly raises the number of initial random conformations and reduces the search depth of each thread”

 to

“Vina-GPU is based on a modified Monte-Carlo using simulating annealing AI algorithm. It greatly raises the number of initial random conformations and reduces the search depth of each thread” in page 1 line 13 (highlighted in yellow).

  1. In Introduction section, page 2, line 47, although the metropolis acceptance rule is a known rule in optimization algorithms, it is suggested to include a reference for metropolis rule.

Response:

Thanks for your advice. We’ve included a reference [17] for metropolis rule in page 2 line 47 (highlighted in yellow).

[17] Metropolis, Nicholas, et al. "Equation of state calculations by fast computing machines." The journal of chemical physics 21.6 (1953): 1087-1092.

  1. In Introduction section, page 5, in equations (5)-(7), what information contains the parameters a and ak?

Response:

Thanks for your advice. We’ve added a description (highlighted in yellow) of the parameters a and ak in Page 6 above Equation 8. The description is as follows (Page 6) ,

And α (αk) means the step size in the direction p (pk), along the decrease of the SF value.

  1. In the Results and Discussion section, page 6, lines 179-190, Tables S1- S3 are cited, but I did not receive the supplementary tables to analyze.

Response:

Thanks for your suggestion. The supplementary materials are included in the new submission, which contains a Vina_GPU_SI.pdf file and a Data S1.xlsx file.

  1. In the Results and Discussion section, page 7, line 207, change “2bm2 and 1jyq converges” by “2bm2 and 1jyq converge”.

Response:

Thanks for your suggestion. We’ve changed “2bm2 and 1jyq converges” by “2bm2 and 1jyq converge” in page 7 line 212 (highlighted in yellow).

  1. In the Results and Discussion section, page 7, lines 219-220, change “the docking score and RMSD converges slowly” by “the docking score and RMSD converge slowly”.

Response:

Thanks for your suggestion. We’ve changed “the docking score and RMSD converges slowly” by “the docking score and RMSD converge slowly” in page 7 line 224 (highlighted in yellow).

  1. In the Results and Discussion section, page 13, in Figure 10, the information included in the rectangle is not readable. This information should be put in a larger size in each graph of Figure 10.

Response:

Thanks for your advice. We’ve enlarged the size of figure and label in Figure 10 (Page 13), and the information becomes more readable. 

  1. In the Conclusion section, page16, line 341, change “tranditional optimizer” by “traditional optimizer”.

Response:

Thanks for your suggestion. We’ve changed “tranditional optimizer” by “traditional optimizer”in page 17 line 353 (highlighted in yellow).

  1. In the Conclusion section, page16, line 342, change “VINA-GPU can updates” by “VINA-GPU can update”.

Response:

Thanks for your suggestion. We’ve changed “VINA-GPU can updates” by “VINA-GPU can update” in page 17 line 354 (highlighted in yellow).

  1. In the Conclusion section, page16, lines 361-362, the sentence “A heuristic function was automatically fitted the most important hyperparameter (search_depth) based on large testing experiments.” should be rewritten for better understanding.

Response:

Thanks for your advice. We’ve changed the sentence “A heuristic function was automatically fitted the most important hyperparameter (search_depth) based on large testing experiments” by A heuristic function was fitted based on large testing experiments to automatically set the most important hyperparameter (search_depth) in page 17 line 363-364 . (highlighted in yellow).

  1. In the Conclusion section, page17, line 365, change “station servers and cloud computations etc.” by “station servers, and cloud computations, etc”.

Response:

Thanks for your suggestion. We’ve changed “station servers and cloud computations etc.” by “station servers, and cloud computations, etc” in page 17 line 367 (highlighted in yellow).

Reviewer 2 Report

Authors reported a method to implement the popular docking software AutoDock Vina with GPUs. The overall study is well performed, detailed and clear. The method proved to be several folds faster than the original approach.

However, there are some issues/clarifications the authors should address:

1) Figure 11b: "Vina-GPU on GPUs" results should be the same as those reported in Figure 5 (Vina-GPU) but it seems they differ (e.g., the RMSD value >8Å shown in Figure 5b is missing in Figure 11b). Please clarify.

2) Supplementary document is in LaTeX, please provide in .pdf or .doc formats.

3) In page 2, lines 56-59 authors reported: "The scale of compounds in virtual screens is vital since the more candidate compounds to be screened,
the lower rate of failure and the more favorable quality of leading compounds can be reached."  Authors should support their statement by providing at least a reference where it is demonstrated that the hit-rate varies with the library size.

4) Authors defined "large" a virtual screening performed on a library of ~9000 compounds. Nowadays, a virtual screen on such a library size is not  considered large, thus I suggest the authors to avoid the word "large" for that case study.

Author Response

Response to Reviewer #2

We would like to thank the reviewer for the valuable insights and suggestions to improve the quality of our manuscript. The following changes have been made to address the Reviewer’s comments, where all the changes are highlighted in yellow in the manuscript.

  1. The Reviewer comments that

Authors reported a method to implement the popular docking software AutoDock Vina with GPUs. The overall study is well performed, detailed and clear. The method proved to be several folds faster than the original approach.

Response:

Thanks for your nice comments.

  1. The Reviewer comments that

1Figure 11b: "Vina-GPU on GPUs" results should be the same as those reported in Figure 5 (Vina-GPU) but it seems they differ (e.g., the RMSD value >8Å shown in Figure 5b is missing in Figure 11b). Please clarify.

Response:

Thanks for your suggestion. In fact, the difference (e.g., the missing part) in the results happens by the randomness of our two experiments of running Vina-GPU on GPU. To keep the consistency of Figure 11b (Page 14) and Figure 5 (Page 9) , we replaced the “Vina-GPU on GPUs” (both RMSD and score) results shown in Figure 11b  (Page 14) with the “Vina-GPU” (both RMSD and score) demonstrated in Figure 5 (Page 9). We also recalculated the Pearson correlation coefficient (0.966). Details can see Figure 11b  (Page 14) in the revised manuscript.

2Supplementary document is in LaTeX, please provide in .pdf or .doc formats.

Response:

Thanks for your suggestion. The Supplementary document (Vina_GPU_SI.pdf) in .pdf format is included in the new submission.

3In page 2, lines 56-59 authors reported: "The scale of compounds in virtual screens is vital since the more candidate compounds to be screened,the lower rate of failure and the more favorable quality of leading compounds can be reached."  Authors should support their statement by providing at least a reference where it is demonstrated that the hit-rate varies with the library size.

Response:

Thanks for your advice. We’ve provided a reference [19] where it is demonstrated that the hit-rate varies with the library size.

[19] Lyu, Jiankun, et al. "Ultra-large library docking for discovering new chemotypes." Nature 566.7743 (2019): 224-229.

4Authors defined "large" a virtual screening performed on a library of ~9000 compounds. Nowadays, a virtual screen on such a library size is not considered large, thus I suggest the authors to avoid the word "large" for that case study.

Response:

Thanks for your advice. We’ve changed the title of section 3.7 from“A Case for Large Virtual Screening”to  “A Case for Virtual Screening”and avoided the word "large" for that case study in section 3.7  (Page 12).

Reviewer 3 Report

This work by Tang and co-workers presented a GPU implementation of the Autodock Vina code based on OpenCL framework. The whole work is in general interesting and would bring an accelerated docking tool, however, the writing of this manuscript is not that satisfactory. 

A few major issues need to be taken care of:

1) The authors described Vina-GPU as a method, but in reality it is a program. 

2) The Broyden-Fletcher-Goldfarb-Shanno (BFGS) method is a very well known optimization method and it has nothing to do with artificial intelligence (AI). 

3) Figure 2 and related description is hard to understand, especially the third panel on the right-hand side.

4) e in Equation 2 should be coined as potential energy rather than free energy

5) The Algorithm 1 scheme after line 171 is unfortunately difficult to understand

6) In general, any GPU code especially in the field of scientific computing should faithfully reproduce the CPU code result, otherwise, the GPU code fails to quality the standard of usefulness. Unfortunately, this Vina-GPU cannot achieve this basic goal. 

7) The authors did not show the RMSD comparison of the 9125 compounds from drugbank (apart from figure 13), is this due to the large error introduced by the GPU code? 

8) The choice of the three drug compounds in Figure 14 is not very good because very few rotatable bonds were involved. 

9) The most severe problem of this Vina-GPU code lies in the tuning of two hyperparameters (GPU threads and depths) which are highly dependent on the protein target. Unfortunately, the authors did not try to make an effort to resolve this issue. 

Author Response

Response to Reviewer #3

We would like to thank the reviewer for the valuable insights and suggestions to improve the quality of our manuscript. The following changes have been made to address the Reviewer’s comments, where all the changes are highlighted in yellow in the manuscript.

  1. The Reviewer comments that

This work by Tang and co-workers presented a GPU implementation of the Autodock Vina code based on OpenCL framework. The whole work is in general interesting and would bring an accelerated docking tool, however, the writing of this manuscript is not that satisfactory.

Response:

Thanks for your nice comments.

  1. The Reviewer comments that

A few major issues need to be taken care of:

  1. The authors described Vina-GPU as a method, but in reality it is a program.

Response:

Thanks for your suggestion. Indeed, Vina-GPU looks more like a program. However, we have modified this method before realizing this program. We greatly raise the number of initial ligand conformation and reasonably reduce the searching steps for each conformation, which is mentioned in page 1 line 14 and demonstrated in Figure 1 (Page 4) and Algorithm 1 (Page 4) . Moreover, we also verify the effectiveness of our improved method Vina-GPU through a large number of experiments (see Figure 5-9, 11-13).

  1. The Broyden-Fletcher-Goldfarb-Shanno (BFGS) method is a very well known optimization method and it has nothing to do with artificial intelligence (AI).

Response:

Thanks for your suggestion. Exactly,the Broyden-Fletcher-Goldfarb-Shanno (BFGS) method is a very well-known optimization method,and it has been widely used in artificial intelligence (AI) algorithms. In order to eliminate its ambiguity, we deleted all descriptions on that BFGS belongs to artificial intelligence (Page 1, Page 3, Page 17; highlighted in yellow).

  1. Figure 2 and related description is hard to understand, especially the third panel on the right-hand side.

Response:

Thanks for your suggestion. To reach a better understanding of the meaning of Figure 2, we've added an example (page 4, line 151-154) and some related descriptions. Details are as follows:

For example, the node 0 has two children-nodes (the node 1 and the node 4) and so the row 0 has two “T”s (indicating “True”) in the 1st and 4th column (Figure 2). Thus, the recursive traverse of the heterogeneous tree can be converted into an iterative traverse of the node list and children map which fits the OpenCL standard.

  1. e in Equation 2 should be coined as potential energy rather than free energy

Response:

Thanks for your suggestion. The description of “free energy” is replaced with “potential energy” (Page 5 above Equation 2, highlighted with yellow).

  1. The Algorithm 1 scheme after line 171 is unfortunately difficult to understand

Response:

Thanks for your advice. We’ve added a detailed description of Algorithm 1 for more clear understanding of our method in page 6 line 171-176 (highlighted in yellow). These descriptions are as follows,

Algorithm 1 proposed the pseudocode of our Vina-GPU.In Algorithm 1, Mutation(.) means a random mutation of the POT in a ligand conformation; BFGS(.) represents the BFGS optimizationmethod which is described in Equation (4)-(7); Scoring(.) is the potential energy of a binding pose described in Equation (2)-(3); Metropolis(.) is the metropolis acceptance criterion described in Equation (8); and Clustering & Sorting (.) is the aggregation and reordering (based on the docking score) of all ligand conformations among all threads.

  1. In general, any GPU code especially in the field of scientific computing should faithfully reproduce the CPU code result, otherwise, the GPU code fails to quality the standard of usefulness. Unfortunately, this Vina-GPU cannot achieve this basic goal.

Response:

Thanks for your suggestion. Maybe caused by our expression problem, we did not emphasize that our Vina-GPU code on GPUs can reproduce very close docking results with the original Autodock Vina code on CPUs(Figure 5, Figure 10, Figure 11 , Figure 13 and Table 1). In our experiments on 140 complexes, the average RMSD of AutoDock Vina and Vina-GPU are 1.5 and 1.7, respectively, and their Pearson correlation coefficient of the scores is 0.965(Figure 5, Figure 10 and Figure 11). In real virtual screening of the Drugbank compound library, our Vina-GPU reproduces the same 14 compounds from the top 15 and the same 46 compounds from the top 50 ones as these by the original Autodock Vina code on the CPU(Table 1, page 16), and the Pearson correlation coefficient of their docking scores is 0.981(Figure 13, page 15). Therefore, our Vina-GPU code on GPUs can reproduce the very close docking results ith the original Autodock Vina code on CPUs, and also can achieve an average of 21 fold and a maximum of 50 fold docking acceleration against the original autodock Vina code on the CPU. In order to objectively emphasize the characteristics of our method, we added some description in the revised draft. Details are as follows (highlighted in yellow),

These results show that our Vina-GPU achieves the very close docking scores with AutoDock Vina on CPU core.  (Page 9)

These results show that our Vina-GPU achieves the similar docking RMSD with AutoDock Vina. (Page 10)

These results show that our Vina-GPU achieves the highly similar docking scores with AutoDock Vina on CPU. (Page 17)

  1. The authors did not show the RMSD comparison of the 9125 compounds from drugbank (apart from figure 13), is this due to the large error introduced by the GPU code?

Response:

Thanks for your suggestion. Indeed, our results did not show the RMSD comparison of the 9125 compounds from Drugbank because there is still no X-ray structure (namely the ground truth structure) of the 9125 compounds binding with our target receptor 1xm6, and it can't calculate the RMSD value. To illustrate the performance of our Vina GPU on RMSD, we calculated the full RMSD comparison on our 140 complexes(Figure5b, page 9).

  1. The choice of the three drug compounds in Figure 14 is not very good because very few rotatable bonds were involved.

Response:

Thanks for your advice. Exactly,the choice of the three drug compounds in Figure 14 (page 15) is not very good because very few rotatable bonds were involved. In this work,Figure 14 (page 15) is just to demonstrate that our Vina-GPU can realize the docking acceleration and achieve close docking results in virtual screening. In our future work, more experiments will be implemented which involves the consideration of compound molecules with more rotatable bonds, and the consider of more compound databanks.

  1. The most severe problem of this Vina-GPU code lies in the tuning of two hyperparameters (GPU threads and depths) which are highly dependent on the protein target. Unfortunately, the authors did not try to make an effort to resolve this issue.

Response:

Thanks for your suggestion. Indeed,the performance of this Vina-GPU code heavily lies on the tuning of two hyperparameters (GPU threads and depths), and also are highly dependent on the protein target. In this work, we have done some efforts to resolve this issue. For instance, we investigated the influence of the two hyperparameters on three typical complexes (small, medium and large) (Figure 3 and 4, page 8). Here, the definition of small, medium and large complexes are decided by the complexity of the ligand rather than protein target. Of course, in the future work, we will further consider the influence of the hyperparameters (GPU threads and depths) on model performance for more levels of complexity about compounds or protein targets. We added some related descriptions in the conclusion section(page 18, highlighted in yellow)as follows, 

Furthermore, we will further consider the influence of the hyperparameters (GPU threads and depths) on model performance for more levels of complexity about compounds or protein targets.

Reviewer 4 Report

The author proposed a new artificial intelligence-based method, AutoDock Vina-GPU, for accelerating AutoDock Vina with GPUs. The results shown in the paper are very impressive. AutoDock Vina-GPU speeds up to around 20 folds compared with CPU version AutoDock Vina and keeps the comparable performance with AutoDock Vina. Overall, the paper is well written, and the result sections are organized very clearly. I only have serval minor comments and hope those suggestions can help to improve the document for publishing.  

Minor 

  1. Figure 1 is slightly confusing. If all MCs in the device part are run parallelly, then the arrow from the “grid cache and random map” box should point to all MC boxes, not only point to MC0. Besides, I would suggest moving the detailed MC sub-figure from the middle to the right because, in the current version, it is not clearly indicated that it is a sub-figure. It would spend readers serval minutes to understand that it is the detailed sub-figure of MC step. 
  2. In Figures 5 and 11 and the corresponding result sections, please provide the average docking score and RMSD in comparison between Vina and Vina-GPU associated with the statistic P-values. Also, show how many good cases (RMSD<2) Vina and Vina-GPU can give.
  3. In Figures 6, 8, and 12, the white dots are too small and not clear. Increase the dot size and use red color or change the dot to a short line. 
  4. In figure 10, the x- and y-axis fonts are too small. 

Author Response

Response to Reviewer #4

We would like to thank the reviewer for the valuable insights and suggestions to improve the quality of our manuscript. The following changes have been made to address the Reviewer’s comments, where all the changes are highlighted in yellow in the manuscript.

  1. The Reviewer comments that

The author proposed a new artificial intelligence-based method, AutoDock Vina-GPU, for accelerating AutoDock Vina with GPUs. The results shown in the paper are very impressive. AutoDock Vina-GPU speeds up to around 20 folds compared with CPU version AutoDock Vina and keeps the comparable performance with AutoDock Vina. Overall, the paper is well written, and the result sections are organized very clearly. I only have serval minor comments and hope those suggestions can help to improve the document for publishing.

Response:

Thanks for your nice comments.

  1. The Reviewer comments that

Minor

  1. Figure 1 is slightly confusing. If all MCs in the device part are run parallelly, then the arrow from the “grid cache and random map” box should point to all MC boxes, not only point to MC0. Besides, I would suggest moving the detailed MC sub-figure from the middle to the right because, in the current version, it is not clearly indicated that it is a sub-figure. It would spend readers serval minutes to understand that it is the detailed sub-figure of MC step.

Response:

Thanks for your advice. In the revised version, we modified Figure 1 (page 4) according to your suggestions to avoid possible confusions.

2In Figures 5 and 11 and the corresponding result sections, please provide the average docking score and RMSD in comparison between Vina and Vina-GPU associated with the statistic P-values. Also, show how many good cases (RMSD<2) Vina and Vina-GPU can give.

Response:

Thanks for your advice. We’ve added the average of docking score and RMSD as well as the number of good cases (RMSD<2) in comparison between Vina and Vina-GPU associated with the statistic P-values in the revised manuscript (highlighted in yellow). These details are as follows,

(1)     Page 9 line 236: The average docking score of AutoDock Vina and Vina-GPU 230are -8.9 and -8.7, respectively

(2)     Page 9 line 243-245: And for Vina-GPU, 107 out of 140 RMSD results are within 2 Å, while 114 out of 140 for AutoDock Vina. The average RMSD of AutoDock Vina and Vina-GPU are 1.5 and 1.7, respectively.

(3)     Page 12 line 309-310: The average docking score of Vina-GPU on CPUs and GPUs are -8.6 and -8.7, respectively.

(4)     Page 12 line 313-315: And for Vina-GPU on CPUs, 104 out of 140 RMSD results are within 2 Å, while 107 out of 140 on GPUs. The average RMSD of Vina-GPU on CPUs and GPUs are 1.8 and 1.7, respectively.

(5)     Page 16 line 332-333: The average docking socre of AutoDock Vina and Vina-GPU are -7.9 and -7.8, respectively

3In Figures 6, 8, and 12, the white dots are too small and not clear. Increase the dot size and use red color or change the dot to a short line.

Response:

Thanks for your advice. We’ve increased the dot size and line width in Figure 6, 8, and 12. The dot color remained white due to the limitation on the color of the python seaborn toolkit, which is used for violin plot demonstration.

4In figure 10, the x- and y-axis fonts are too small.

Response:

Thanks for your advice. We’ve enlarged the axis size in Figure 10 (page 13).

Round 2

Reviewer 2 Report

The manuscript looks ready for being published on Molecules.

Author Response

Thanks for your nice comments.

Reviewer 3 Report

The authors addressed most of my questions in the first review, however, point 8 was not addressed. Some new example ligands with more rotatable bonds are expected to replace the existing ones. 

Author Response

Thanks for your nice comments. Based on your and editor’s suggestions, and to eliminate the misunderstanding of "point 8: some new example ligands with more rotatable bonds", we have implemented the complete removal of Figure 14 and its relevant descriptions from the manuscript in the revised version.